# A LENNARD-JONES LAYER FOR DISTRIBUTION NORMALIZATION

## ABSTRACT

We introduce a *Lennard-Jones layer* (LJL) to equalize the density across the distribution of 2D and 3D point clouds by systematically rearranging points without destroying their overall structure (*distribution normalization*). LJL simulates a dissipative process of repulsive and weakly attractive interactions between individual points by considering the nearest neighbor of each point at a given moment in time. This pushes the particles into a potential valley, reaching a well-defined stable configuration that approximates an equidistant sampling after the stabilization process. We apply LJLs to redistribute randomly generated point clouds into a randomized uniform distribution. Moreover, LJLs are embedded in point cloud generative network architectures by adding them at later stages of the inference process. The improvements coming with LJLs for generating 3D point clouds are evaluated qualitatively and quantitatively. Finally, we apply LJLs to improve the point distribution of a score-based 3D point cloud denoising network. In general, we demonstrate that LJLs are effective for distribution normalization which can be applied at negligible cost without retraining the given neural networks.

## 1 INTRODUCTION

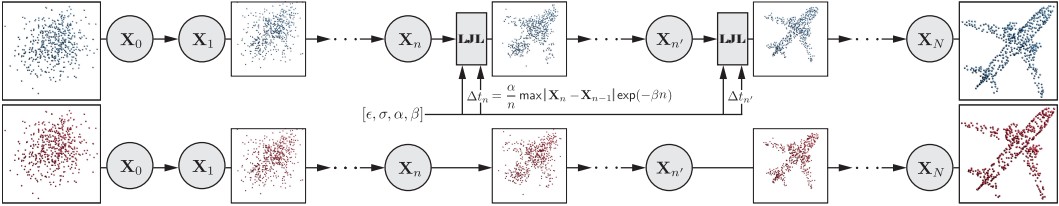

Figure 1: An overview of the integration of LJLs into the inference process of well-trained generative models. A well-trained model generates a meaningful point cloud from random noise in a sequential way (bottom row). LJLs are inserted after certain intermediate-generation steps with damping step size $\Delta t_n$ and LJ potential parameters $\epsilon$ and $\sigma$ (top row). Embedding LJLs in generative models can improve the generation results with normalized point distribution.

Next to triangle meshes, point clouds are one of the most commonly used representations of 3D shapes. They consist of an unordered set of 3D points without connections and are the native output format of 3D scanners. They also map well to the structure of neural networks, either in generative or processing (e.g. denoising) tasks. A major problem with point clouds is, that their entropy heavily depends on the distribution of points on the surface. In areas with holes, not enough information about the shape is expressed, while clustered areas waste storage without adding additional details. We introduce a so-called *Lennard-Jones layer*(LJL) for the distribution normalization of point clouds. We use the term *distribution normalization* to describe the process of systematically rearranging the points' positions in order to equalize their density across the surface without changing the overall shape. More formally, *distribution normalization* describes the property of maintaining a global structure while maximizing minimum distances between points. The Lennard-Jones(LJ) potential is widely considered an archetype model describing repulsive and weakly attractive interactions of atoms and molecules. It was originally introduced by John Lennard-Jones (Jones, 1924a;b; Lennard-Jones, 1931) who is nowadays recognized as one of the founding fathers of computational chemistry. Applications of LJ-based numerical simulations can be found in different

scientific communities ranging from molecular modelling (Jorgensen et al., 1996) and soft-matter physics (Al-Raeei & El-Daher, 2019) to computational biology (Hart & Istrail, 1997). For a given point cloud, we aim for a transformation from the initial distribution into a blue noise-type arrangement by simulating the temporal evolution of the associated dynamical system. Each point of the point cloud is interpreted as a particle in a dynamic scene to which the LJ potential is applied as a moving force. In each iteration of the temporal integration process, the point cloud is decomposed into a new set of independent subsystems, each containing a single pair of particles. Each of these subsystems is then simulated individually, which greatly increases stability. As we also include dissipation into our system, the point configuration of the particle system will be stabilized and form randomized uniform point distribution after a sufficient number of iterations. Learning-based point clouds related research has been explored for years, especially in the fields of generation and denoising. Existing models are solely trained to generate and denoise the point cloud without considering the overall point distribution and therefore exhibit the common problems of holes and clusters. By embedding LJLs into well-trained architectures, we are able to significantly improve their results in terms of point distribution while at the same time introducing minimal distortion of the shape. We also circumvent resource- and time-intensive retraining of these models, as LJLs are solely embedded in the inference process. This way, LJLs are a ready-to-use plug-in solution for point cloud distribution optimization.

## 2 RELATED WORK

**Molecular Dynamics**   Simulating molecular dynamics is an essential tool for applications such as understanding protein folding (Creighton, 1990) or designing modern drugs (Durrant & McCammon, 2011) and materials (Van Der Giessen et al., 2020). While the motion of atoms and molecules can in principle be obtained by solving the time-dependent Schrödinger equation, such a quantum mechanics approach remains too computationally prohibitive in practice to investigate large molecular systems. Instead, classical molecular dynamics uses Newtonian mechanics treating the nuclei as point particles in a force field that accounts for both, their mutual as well as electronic interactions. This force field is derived from a potential energy function (e.g. the LJ potential) that is formulated either from expectation values of the quantum system or using empirical laws (Rapaport, 2004). Given the typically large number of atoms and molecules involved in molecular dynamics, an analytical solution of the resulting Newtonian mechanical system is usually out of reach. Consequently, numerical methods that evaluate the position of each nucleus at (fixed or adaptive) time intervals are used to find computational approximations to the solutions for given initial conditions (Hochbruck & Lubich, 1999). Classical molecular dynamics simulators are *LAMMPS* Molecular Dynamics Simulator (2014), which utilizes "velocity Verlet" integration (Verlet, 1967), *RATTLE* (Andersen, 1983) and *SHAKE* (Ryckaert et al., 1977), in which the strong covalent bonds between the nuclei are handled as rigid constraints. Modern approaches allow for the more efficient numerical simulation of large molecular structures as efficiency has further increased, either through algorithmic improvements (Michels & Desbrun, 2015) or by leveraging specialized hardware for parallel computing (Walters et al., 2008). Among others, Alharbi et al. (2017) provided a *Eurographics* tutorial on real-time rendering of molecular dynamics simulations. More recent work also addresses immersive molecular dynamics simulations in virtual (Bhatia et al., 2020; Jamieson-Binnie et al., 2020) and augmented reality (Eriksen et al., 2020).

**Blue Noise Sampling**   Colors of noise have been assigned to characterize different kinds of noise with respect to certain properties of their power spectra. In visual computing, blue noise is usually used more loosely to characterize noise without concentrated energy spikes and minimal low-frequency components. Due to its relevance in practical applications such as rendering, simulation, geometry processing, and machine learning, the visual computing community has devised a variety of approaches for the generation and optimization of blue noise. Several methods can maintain the Poisson-disk property, by maximizing the minimum distance between any pair of points (Cook, 1986; Lloyd, 1982; Dunbar & Humphreys, 2006; Bridson, 2007; Balzer et al., 2009; Xu et al., 2011; Yuksel, 2015; Ahmed et al., 2017). Schlömer et al. (2011) introduced farthest-point optimization(FPO) that will maximize the minimum distance of the point set by iteratively moving every point to the farthest distance from the rest of the point set. de Goes et al. (2012) successfully generated blue noise through optimal transport(BNOT). Their method has widely been accepted as the benchmark for best-quality blue noise, which was recently surpassed by Gaussian blue noise(GBN) (Ahmed et al., 2022). GBN is classified as kernel-based methods (Öztireli et al., 2010;

Fattal, 2011; Ahmed & Wonka, 2021) that augment each point with a kernel to model its influence. There are several previous works that we consider as methodologically closely related to ours as the authors applied the physical-based particle simulation to synthesize blue noise. The method proposed by Jiang et al. (2015), is based on smoothed-particle hydrodynamics(SPH) fluid simulation which takes different effects caused by pressure, cohesion, and surface tension into account. This results in a less puristic process compared to our work. Both Schmaltz et al. (2010) and Wong & Wong (2017) employ electrical field models, in which electronically charged particles are forced to move towards an equilibrium state after numerical integration because of the electrostatic forces. Adaptive sampling can be achieved by assigning different amounts of electrical charge to particles.

**Generative Models for 3D Point Cloud** Research related to 3D point clouds has recently gained a considerable amount of attention. Some methods aim to generate point clouds to reconstruct 3D shapes from images (Fan et al., 2016) or from shape distributions (Yang et al., 2019). Achlioptas et al. (2017) propose an autoencoder for point cloud reconstruction, while others aim to solve point cloud completion tasks based on residual networks (Xie et al., 2020), with multi-scale (Huang et al., 2020) or cascaded refinement networks (Wang et al., 2020) or with a focus on upscaling (Li et al., 2019). Some methods solve shape completion tasks by operating on point clouds without structural assumptions, such as symmetries or annotations (Yuan et al., 2018), or by predicting shapes from partial point scans (Gu et al., 2020). For 3D shape generation methods either use point-to-voxel representations (Zhou et al., 2021) or rely on shape latent variables to directly train on point clouds (Luo & Hu, 2021a; Zeng et al., 2022), by either employing normalizing flows (Rezende & Mohamed, 2015; Dinh et al., 2017) or point-voxel convolutional neural networks (Liu et al., 2019). ShapeGF (Cai et al., 2020) is an application of score-based generative models (Song & Ermon, 2019) for 3D point cloud generation, which uses estimated gradient fields for shape generation where point cloud is viewed as samples of a point distribution on the underlying surface. Therefore, with the help of stochastic gradient ascent, sampled points are moving toward the regions near the surface. Luo & Hu (2021b) first attempt to apply denoising diffusion probabilistic modeling(DDPM) (Sohl-Dickstein et al., 2015; Ho et al., 2020) in the field of point cloud generation. Points in the point cloud are treated as particles in the thermodynamic system. During the training process, the original point cloud is corrupted by gradually adding small Gaussian noise at each forward diffusion step, while the network is trained to denoise and reverse this process.

**Denoising Techniques for 3D Point Cloud** Approaches for denoising point clouds aim to reduce perturbations and noise in point clouds and facilitate downstream tasks like rendering and remeshing. Similar to images, denoising for point clouds can be solved with linear (Lee, 2000) and bilateral (Fleishman et al., 2003; Digne & de Franchis, 2017) operators, or based on sparse coding (Avron et al., 2010). Although these methods can generate less noisy point clouds, they also tend to remove important details. Duan et al. (2018) leverages the 3D structure of point clouds by using tangent planes at each 3D point to compute a denoised point cloud from the weighted average of the points. Many existing approaches employ neural network architectures for point cloud denoising purposes, by building graphs or feature hierarchies (Pistilli et al., 2020; Hu et al., 2020), by employing differentiable rendering for point clouds (Yifan et al., 2019), or by classifying and rejecting outliers (Rakotosaona et al., 2020). Hermosilla et al. (2019) propose an unsupervised method for denoising point clouds, which combines a spatial locality and a bilateral appearance prior to mapping pairs of noisy objects to themselves. The majority of learning-based denoising methods directly predict the displacement from noisy point positions to the nearest positions on the underlying surface and then perform reverse displacement in one denoising step. Luo & Hu (2021c) propose an iterative method where the model is designed to estimate a score of the point distribution that can then be used to denoise point clouds after sufficient iterations of gradient ascent. More specifically, clean point clouds are considered as samples generated from the 3D distribution $p$ on the surface, while noise is introduced by convolving $p$ with noise mode $n$. The score is computed as the gradient of the log-probability function $\nabla \log(p \cdot n)$. They claim that the mode of $p \cdot n$ is the ground truth surface, which means that denoising can be realized by converging samples to this mode.

## 3 LENNARD-JONES LAYER

The Lennard-Jones(LJ) potential $V$ is defined as

$$V(r) = 4\epsilon \left( \left( \frac{\sigma}{r} \right)^{12} - \left( \frac{\sigma}{r} \right)^{6} \right) , \tag{1}$$

which is a function of the distance between a pair of particles simulating the repulsive and weakly attractive interactions among them. The distance $r$ is measured by the Euclidean metric. The first part $(\cdot)^{12}$ attributes the repulsive interaction while the second part $(\cdot)^6$ attributes the soft attraction. With the help of the parameters $\epsilon$ and $\sigma$, the LJ potential can be adjusted to different use cases. Fig 2 (left part) illustrates the role of these parameters: The potential depth is given by the parameter $\epsilon > 0$ which determines how strong the attraction effect is, and $\sigma$ is the distance at which the LJ potential is zero.

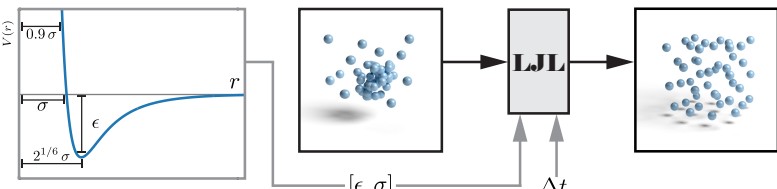

Figure 2: Illustration of the Lennard-Jones(LJ) potential and the Lennard-Jones layer (LJL). The strength and position of the repulsive and weakly attractive zones in the LJ potential are controlled by the hyperparameters $\epsilon$ and $\sigma$, while $\Delta t$ controls the damping step size of the LJL. Applying the LJL to a point cloud leads to a more uniform distribution of the points.

The corresponding magnitude of LJ force can be obtained by taking the negative gradient of Eq. (1) with respect to $r$ which is $F_{LJ}(r) = -\nabla_r V(r)$. The direction of LJ force on each particle is always along the line connecting the two particles, which point towards each other when two particles are attracted and vice versa. Equilibrium distance $r_E$ is where $F_{LJ}(r_E) = 0$ and the LJ potential reaches the minimum value. For the distance $r < r_E$, the repulsive part dominates, while for $r > r_E$ the particles start to attract to each other. When $r$ gets bigger, the attractive force will decrease to zero, which means that when two particles are far enough apart, they do not interact. If we assume a uniform mass distribution among all particles and normalize with the mass, the corresponding equations of motion acting on a pair of particles whose positions are described by $x_1, x_2 \in \mathbb{R}^2$ or $\mathbb{R}^3$ can be obtained by

$$r = |x_1 - x_2|, \quad \ddot{x}_1 = F_{LJ}(r) \cdot \frac{(x_1 - x_2)}{r}, \quad \ddot{x}_2 = F_{LJ}(r) \cdot \frac{(x_2 - x_1)}{r}.$$

Given a 2D or 3D input point cloud in Euclidean space, we aim for a transformation from a white noise- into a blue noise-type arrangement by simulating the temporal evolution of the associated dynamical system. In this regard, we use a formulation of LJ potential where pairs of particles are grouped as a pair potential to simulate the dynamic procedure which is called Lennard-Jones Layer (LJL). The computations within LJL are summarized in Alg. 1. Within a system containing a number of particles, the potential energy is given by the sum of LJ potential pairs. This sum shows several local minima, each associated with a low energy state corresponding to the system's stable physical configuration. It is an essential aspect of our method, that we do not compute the complete numerical solution of the resulting $N$-body problem (in which $N$ denotes the number of involved particles). We only take into account the closest neighbor ($k = 1$ neighborhood) of each particle within every iteration of the temporal integration process. Note that being the closest neighbor of each particle is not a mutual property. Therefore, the potential energy of each pair of interactions has only a single energy valley. This strategy is equivalent to the decomposition of the particle system into several independent subsystems in which every subsystem just contains a single pair of particles. In each iteration, different subsystems (i.e., new pairs) are formed. As the subsystems are not interacting within each iteration, the sum of the LJ potentials has just a single (global) minimum. LJL creates pairwise independent subsystems using *nearest neighbor search* (NNS) by assigning the nearest points to the current point set $X_i$. We simulate this process by repetitively integrating forward in time using a decaying time step $\Delta t$ which dampens exponentially according to

$$\Delta t(\cdot) = \alpha \, \mathsf{exp}(-\beta\,(\cdot)), \tag{2}$$

in which $\alpha > 0$ refers to the initial step size, and $\beta > 0$ defines the damping intensity. For each pair, the position update is then computed independently by integrating the equation of motion numerically using a basic Störmer–Verlet scheme. Within this numerical integration process, we intentionally do not make use of the particles' velocities from the previous step as this corresponds to a velocity reset (setting all velocities identically to zero) causing a desired dissipation effect for the final convergence. Moreover, as $V(r)$ increases rapidly when $r < \sigma$, we clamp the LJ potential

from $r = 0.9\sigma$ to $r = 100\sigma$ and furthermore employ the hyperbolic tangent as an activation function restricting the gradient $\nabla_r V(r)$ to the interval $[-1, 1]$ to avoid arithmetic overflows. To this end, we can redistribute any randomly generated point cloud into a uniform distribution by iteratively applying LJLs. This process stops as soon as a stable configuration is reached, which means that the difference between the previous and current generated point clouds is below the tolerated threshold. Consequently, this spatial rearrangement of particles prevents the formation of clusters and holes.

---

**ALGORITHM 1:** The *Lennard-Jones layer*(***LJL***). NNS denotes the *nearest neighbor search*

**Input:** Point cloud $\boldsymbol{X}_i$ and point cloud $\text{NNS}(\boldsymbol{X}_i)$
**Output:** Point cloud $\boldsymbol{X}_{i+1}$.

1    $t_i \leftarrow \Delta t(i)$
2    $r \leftarrow \min\{\max\{|\boldsymbol{X}_i - \text{NNS}(\boldsymbol{X}_i)|, 0.9\,\sigma\}, 100\sigma\}$
3    $\Delta x \leftarrow \tanh(-\nabla_r V(r)) \cdot t_i^2 \,/\, 2$
4    $\boldsymbol{X}_{i+1} \leftarrow \boldsymbol{X}_i + \Delta x \cdot (\boldsymbol{X}_i - \text{NNS}(\boldsymbol{X}_i))/|\boldsymbol{X}_i - \text{NNS}(\boldsymbol{X}_i)|$

---

## 4   Applications

### 4.1   Numerical Examples on 2D Euclidean Plane

The LJ potential itself has two parameters: The repulsion distance $\sigma$ and the attraction strength $\epsilon$. Similar to the time step $\Delta t$, $\epsilon$ scales the gradient, however, it is applied before computing the tanh-function (see Alg. 1). In practice, we find that setting $\epsilon = 2$ results in a well-behaving LJ gradient. The repulsion distance $\sigma$ corresponds to the optimal distance that all particles strive to have from their neighbors. To derive the proper value for $\sigma$, we take inspiration from the point configuration of the hexagonal lattice. The largest minimum distance $r$ between any two points is given by $r = \sqrt{2/(\sqrt{3}\,N)}$ for $N$ points over a unit square (Lagae & Dutré, 2005). Fig. 3 shows the result of different values of $\sigma$ under identical initial conditions where points are initialized in a unit square. An appropriate $\sigma$ results in a uniform point distribution that does not exceed the initial square too much. $\sigma' = \sqrt{2/(\sqrt{3}\,N)}$ denotes the estimated optimal value of $\sigma$. Small $\sigma$'s lack redistribution ability over the point set as they lead to a limited effect of LJL on particles. Given a large $\sigma$, particles tend to interact with all the points nearby and are spread out of the region excessively. After exploring the effect of $\sigma$ on the point distribution over the unbounded region, we introduce fixed boundary conditions where points will stop at the boundaries when they tend to exceed. In this constraint scenario, LJL is robust with respect to $\sigma$ as long as it is sufficiently large. When it comes to the damping step size $\Delta t$ in Eq. 2, we want LJL updates with decreasing intensity controlled by the damping parameters $\alpha$ and $\beta$ in order to converge samples in later iterations. Empirically, we set $\alpha = 0.5$ and $\beta = 0.01$ to ensure a proper starting step size and slow damping rate. More experimental details are in Appendix A.1. We also utilize LJL to redistribute point clouds over mesh surfaces, see Appendix A.2.

**Blue Noise Analysis**   For many applications in visual computing, blue noise patterns are desirable. Due to the fact that LJLs have the ability to normalize the point distribution, we apply LJLs to redistribute samples to randomized uniform distribution solely from white noise distribution and analyze the corresponding blue noise properties. Here we synthesize blue nose distribution in the 2D unit square with periodic boundaries. Inside the region, initial random points are generated and iteratively rearranged by LJLs to generate blue noise. The classic spectral analysis for blue noise includes the power spectrum which averages the periodogram of distribution in the frequency domain and two corresponding 1D statistics: radial power and anisotropy. Radial power averages the power spectrum over different radii of rings and the corresponding variance over the frequency rings is anisotropy indicating regularity and directional bias. We use the point set analysis tool PSA (Schlömer & Deussen, 2011) for spectral evaluation of different 2D blue noise generation methods (1024 points) in Fig. 4.

### 4.2   LJL-embedded Deep Neural Networks

Clusters and holes are consuming the limited number of points without contributing additional information to the result. In the following two applications, LJLs are plugged into the 3D point cloud generation and denoising networks that work in an iterative way (e.g. Langevin dynamics). With the help of LJLs, we can enforce points to converge to a normalized distribution by embedding LJLs

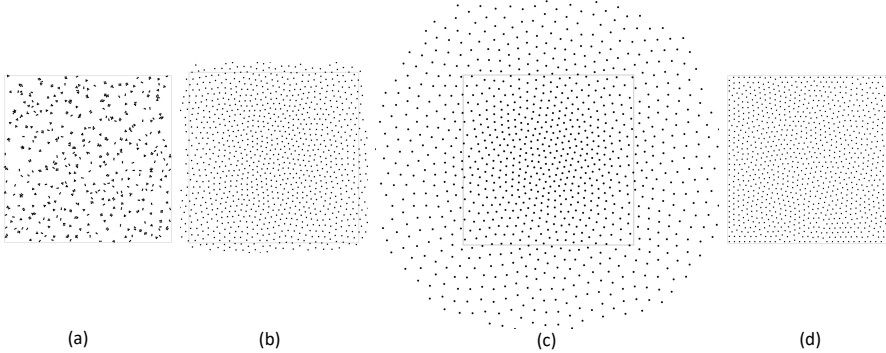

Figure 3: Influence of the parameter $\sigma$. The rectangle denotes the unit square in which 1024 points are initialized. (a) $\sigma = 0.2\sigma'$ leads to clusters; (b) $\sigma = \sigma'$ distributes points well and keeps them close to the boundary; (c) $\sigma = 10\sigma'$ spreads points out of the boundary extremely; (d) $\sigma = 10\sigma'$ with fixed boundary conditions. $\sigma'$ denotes the estimated optimal value of $\sigma$.

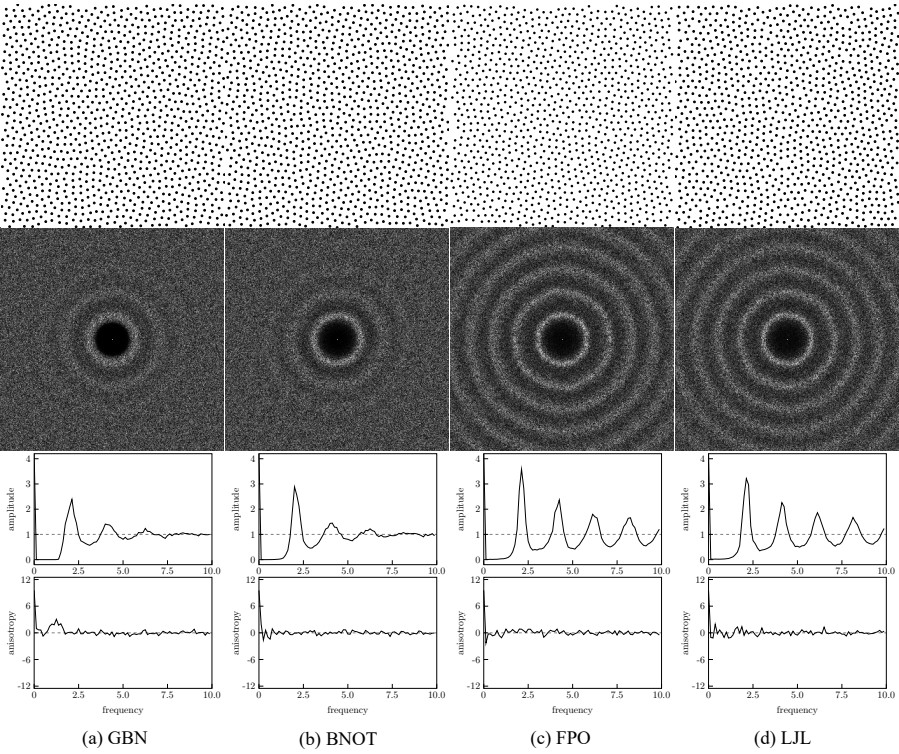

Figure 4: Spectral analysis of different blue noise generation methods.

into refinement steps (generation and denoising). Along with the evaluation metric point-to-mesh distance (*noise score*), we use the mean value of Euclidean distances from each point to its nearest neighbor (Schlömer et al., 2011) denoted as *DistanceScore* to evaluate the point distribution of $N$ points in Eq. 3. We want LJL-embedded results to have as small as possible noise score increments while having large *DistanceScore* increments.

$$\text{DistanceScore} = \frac{1}{N} \sum_{i=0}^{N} |\boldsymbol{X_i} - \text{NNS}(\boldsymbol{X_i})|. \tag{3}$$

**LJL-embedded Generative Model for 3D Point Cloud**   In this section, we show that LJLs can be applied to improve the point distribution of point cloud generative models ShapeGF (Cai et al., 2020) and DDPM (Luo & Hu, 2021b) without the need to retrain them. It has been proven that

the aforementioned two types of probabilistic generative models are deeply correlated (Song et al., 2021). The inference process of both models can be summarized as an iterative refinement process shown in the bottom row of Fig. 1. Initial point cloud $X_0$ is sampled from Gaussian noise $N(0, I)$. The recursive generation process produces each intermediate point cloud $X_n$ according to the previously generated point cloud $X_{n-1}$. Even though there is randomness introduced in each iteration to avoid local minima, the generation process depends heavily on the initial point positions. Every point moves independently without considering the neighboring points, thus generated point clouds tend to form clusters and holes. Fig. 1 provides an overview of the integration of LJLs into the inference process of well-trained generative models. Since the sampling procedure of these generative models is performed recursively, one can insert LJLs into certain intermediate steps and guide the generated point cloud to have a normalized distribution. The generator and LJLs have different goals: the former aims to converge all points to a predicted surface ignoring their neighboring distribution, which is also can be seen as a denoising process. While the latter aims to prevent the formation of clusters and holes to improve the overall distribution. It is therefore important to observe whether LJLs are inhibiting the quality of the generation process. The trade-off between distribution improvement and surface distortion needs to be measured.

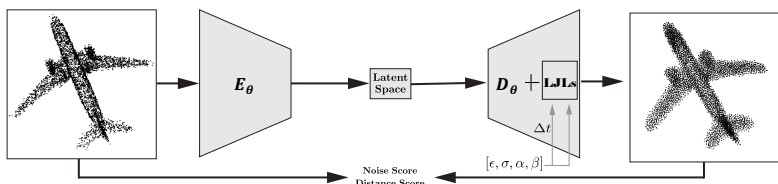

Figure 5: A pipeline for searching LJL parameters through a well-trained autoencoder-decoder.

In order to find LJL parameters that achieve a favorable trade-off, we apply LJLs in the well-trained autoencoder-decoder models. In Fig. 5, given a point cloud $Y$, the autoencoder $E_\theta$ encodes it into latent code $Z$, then the decoder $D_\theta$ which is the generator will utilize iterative sampling algorithms to reconstruct the point cloud conditioned on $Z$. we fix encoder $E_\theta$ and insert LJLs solely in decoder $D_\theta$. To determine proper parameters for LJLs in generation tasks, the reconstructed results need to have a good point distribution and preserve the original shape structures according to *distance* and *noise scores*. The computation related to this task is summarized in Alg. 2.

---

**ALGORITHM 2:** *LJL parameter searching via autoencoder $E_\theta$ and decoder $D_\theta$.*

**Input:** Point cloud $Y$, starting $SS$ and ending $T' < T$ steps.
**Output:** Reconstructed point cloud $X_T$.

1   $Z \leftarrow E_\theta(Y)$
2   $X_0 \sim N(0, I)$
3   **FOR** $t \leftarrow 1...T$
4      **IF** $SS \leq t \leq T'$ :
5         $X_t \leftarrow D_\theta(X_{t-1}, Z)$
6         $X_t \leftarrow LJLs(X_t, \mathsf{NNS}(X_t))$
7      **ELSE**:
8         $X_t \leftarrow D_\theta(X_{t-1}, Z)$
9   **RETURN** $X_T$

---

During the generation process, LJLs are activated from *starting steps($SS$)* till *ending steps($T'$)*. It is not necessarily advantageous to perform LJLs at every iteration, since LJLs slow down the convergence of the generation in the beginning steps. LJL-embedding starts in the second half of the generation steps. We also circumvent embedding LJLs in the last few steps to ensure that no extra noise is introduced. Point clouds are normalized into the cube $[-1, 1]^3$, and we set $\epsilon = 2$ and $\sigma = 5\sigma'$ due to the additional third dimension. The decaying time step $\Delta t$ in $i$-th generation step is set as

$$\Delta t(i) = \frac{\alpha}{i} \mathsf{max}|X_i - X_{i-1}| \mathsf{exp}(-\beta\, i),$$

where $\Delta t(i)$ is scaled by the maximum difference between the current $X_i$ and the previous $X_{i-1}$ point cloud, such that each position modification $\Delta X$ caused by LJLs maintains similar scale. Furthermore, $\Delta t(i)$ is divided by the iteration number to diminish the redistribution effect as generation steps increase. In order to determine $\alpha$ and $\beta$, we perform a systematic parameter searching based on *distance* and *noise scores*. We found that $\alpha = 2.5$ and $\beta = 0.01$ meet the requirements of less noise and better distribution. Incorporating the optimal parameters, we tested LJL-embedded generative

models on ShapeNet (Chang et al., 2015) and found a $99.2\%$ increase in the *DistanceScore* and a $7.8\%$ increase of the noise score for the car dataset. The point distribution is improved by $42.8\%$ while the noise score is only increased by $2.8\%$ for airplanes. Finally, we get a $207.9\%$ increase in *DistanceScore*, and $19.1\%$ increase in the noise score for the chair dataset, shown in Table 1. More details about LJL parameter settings and generation results can be found in the Appendix B.1.

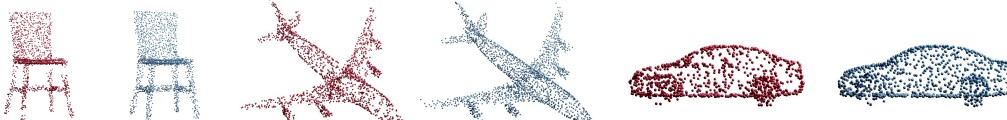

Figure 6: Comparison of generation-only (red) and LJL-embedded generation (blue).

Table 1: *Noise score* and *DistanceScore* increments introduced by LJL-embedded generation.

|  | Noise | Distance |
| --- | --- | --- |
| Airplane | 2.8%↑ | 42.8% ↑ |
| Chair | 19.1% ↑ | 207.9% ↑ |
| Car | 7.8% ↑ | 99.2% ↑ |

Table 2: *Noise score* and *DistanceScore* increments introduced by LJL-embedded denoising.

|  | Noise | | Distance | |
| --- | --- | --- | --- | --- |
|  | Less noise | More noise | Less noise | More noise |
| IT 30 | 0.093% ↑ | 2.63% ↑ | 8.79% ↑ | 10.71% ↑ |
| IT 60 | 0.42% ↓ | 1.37% ↑ | 13.04% ↑ | 16.85% ↑ |

**LJL-embedded Denoising Model for 3D Point Cloud** Given a noisy point cloud $\boldsymbol{X^0} = \{\boldsymbol{x}_i^N\}$ which is normalized into the cube $[-1, 1]^3$, the score-based model Luo & Hu (2021c) utilizes stochastic gradient ascent to denoise it step by step. In each intermediate step, the denoising network $\boldsymbol{S}_\theta(\boldsymbol{X})$ predicts the gradient of the log-probability function (score) which is a vector field depicting vectors pointing from each point position towards the estimated underlying surface. The gradient ascent denoising process will iteratively update each point position to converge to the predicted surface, where we can obtain the denoised result $\boldsymbol{X}^T$. The denoising step can be described as follows:

$$\boldsymbol{X}^{(t)} = \boldsymbol{X}^{(t-1)} + \gamma_t \boldsymbol{S}_\theta(\boldsymbol{X}^{(t-1)}), \ \ t = 1, \ldots, T,$$

where $\gamma_t$ is denoising step size for $t$-th iteration. LJLs are inserted after each intermediate denoising step except the last few iterations to avoid extra perturbation to the denoised results. LJL parameter settings and evaluation metrics used in this task are inherited from previous generation tasks, where $\sigma = 5\sigma'$ and $\epsilon = 2$. The only difference is that the position update scale ($\max|\boldsymbol{X}_i - \boldsymbol{X}_{i-1}|$) for the denoising step is different from that in the generation task. Thus corresponding LJL damping step size $\Delta t$ needs to be determined. We keep $\beta = 0.01$ and search for $\alpha$ by finding the minimum ratio of *noise and distance score increment rate*, as shown in Fig. 8(a). We choose $\alpha = 0.3$ for the following task. To this end, the denoising results not only converge to the predicted surface with less distortion but also distribute uniformly as shown in Fig. 7.

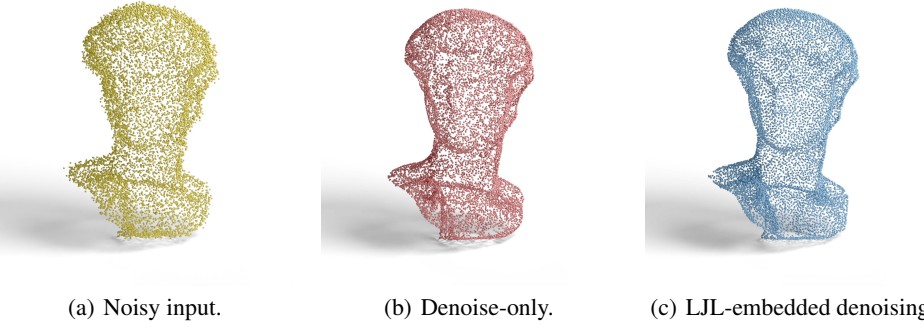

(a) Noisy input.      (b) Denoise-only.      (c) LJL-embedded denoising.

Figure 7: Comparison between denoising without and with LJLs.

We continue to determine appropriate denoising iterations and the performance of denoising models (with and without LJLs) on different noise scales (less and more). This is because denoising tends to progressively smooth the surface and wash out the details. This phenomenon is shown in Fig. 8(b)

and Fig. 8(c). It is clear that no matter the noise scales, for the denoise-only model, *noise score* increases when the number of iterations go up. While for the LJL-embedded denoising model, *noise scores* remain steady for the first 60 denoising iterations, *DistanceScore* of both models decrease when iteration times increase regardless of noise scales. We further evaluate the LJL-embedded denoising model on the test set from Luo & Hu (2021c) shown in Table 2. By combining LJLs with the denoising network, we find that it improves *DistanceScore* with negligible increments of the *noise score*. Moreover, we boost the performance of existing denoising models without extra training. More LJL-embedded denoising results and evaluations are shown in the Appendix B.2.

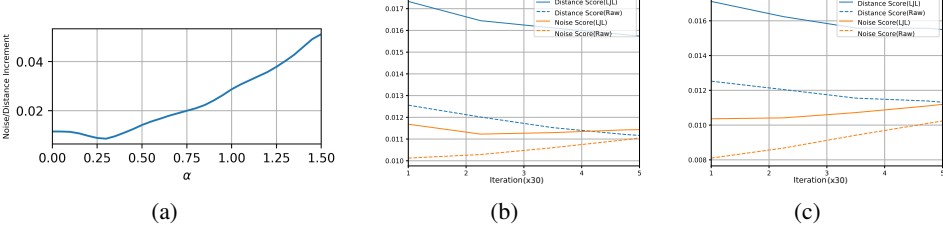

(a)  (b)  (c)

Figure 8: (a) Ratio of *noise and distance score increment rate*. Parameter searching for optimal $\alpha$ by finding the global minima at $\alpha = 0.3$. (b) Denoising results of lower noise scale point clouds with respect to increasing denoising iterations. (c) Denoising results of higher noise scale point clouds with respect to increasing denoising iterations.

## 5 CONCLUSION

We extend the concept of Lennard-Jones potential describing pairwise repulsive and weakly attractive interactions of atoms and molecules to a variety of tasks that require equalizing the density of a set of points without destroying the overall structure of point sets. The dynamic rearrangement of point distribution based on a system of pairwise LJ potentials is called *Lennard-Jones layer* (LJL). LJLs are conceptually and practically capable of generating high-quality blue noise distributions. To demonstrate the benefit of applying LJLs in the context of the 3D point cloud generation task, we incorporate LJLs into generative models ShapeGF (Cai et al., 2020) and DDPM (Luo & Hu, 2021b), in order to generate point clouds with better point distribution. By embedding LJLs in certain intermediate-generation steps, the clusters and holes are decreased significantly because LJLs increase the chances of points converging to cover the entire shape. Finally, we combine LJLs with score-based point cloud denoising network (Luo & Hu, 2021c) such that the noisy point clouds are not only moved toward the predicted surfaces but also maintain uniform distribution. Consequently, this rearrangement prevents the formation of clusters and holes. Since adding LJLs in these cases does not require retraining the network, the cost of applying LJLs is negligible.

The presented work offers several avenues for future work. From an algorithmic perspective, the $k$-nearest neighbor classification performance could be improved through supervised metric learning methods such as neighborhood components analysis and large margin nearest neighbor. The distribution normalization on surfaces could be addressed by means of considering tangential bundles. By doing this, we force particles not to leave tangential manifolds during LJL interactions and in a perfect case to move solely along the geodesics of the implied surfaces. This would require that we do not distort the underlying surfaces while performing distribution normalization. Geodesical distances between points on surfaces also refer to a uniform sampling, which seems more adequate than Euclidean distances of embedding spaces, the reconstruction of geodesics from point clouds has been explored by Crane et al. (2013). From an analytical point of view, the presented LJL for distribution normalization has a discrete, piecewise character as it takes only the closest neighbors into account which change from iteration to iteration. Most common approaches in theoretical mechanics aim for a continuous description, e.g., based on Hamiltonians and their analytical solution or numerical integration. It is theoretically not yet clear if it is possible to obtain a blue noise-type arrangement of particles by means of pure Hamiltonian mechanics. For future work, we would like to further investigate this question. On a slightly different trajectory, we would like to explore the possibilities of relativistically moving particles. If such particles move within a medium with a suitable refraction index, then the power density of radiation grows linearly with frequency. This effect is well studied in the literature and known as Vavilov-Cherenkov radiation (Cherenkov, 1934; Jackson, 1999).

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

APPENDIX

## A   PARAMETER CONFIGURATION OF LENNARD-JONES LAYER

We investigate the behavior of LJL in detail by evaluating the point configuration of point sets on 2D Euclidean plane and 3D mesh surfaces and propose strategies for LJL parameter searching for practical applications.

### A.1   NUMERICAL EXAMPLES ON 2D EUCLIDEAN PLANE

Before applying LJLs to practical applications, we now analyze the behavior of LJLs in different situations (i.e., different numbers of neighbors $k$, with or without attraction term in LJ potential). We set LJL parameters $\alpha = 0.5$, $\beta = 0.01$, $\epsilon = 2$, and $\sigma = 5\sigma'$. All different cases are initialized with uncorrelated randomly distributed points inside a unit square region.

$k$-**NN**   The interaction acting on individual particles according to LJ potential can be summed with a varying number of nearest neighbors. Our theoretical finding guarantees convergence only for the consideration of a single nearest neighbor. The experiment shown in Fig. 9 indicates point configurations of LJLs considering a different number of nearest neighbors $k$. Cases with more than one neighbor increase the difficulty of reaching uniform point distribution.

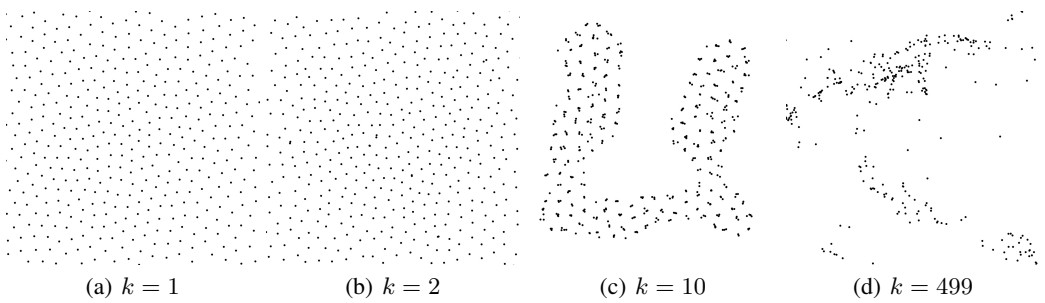

(a) $k = 1$       (b) $k = 2$       (c) $k = 10$       (d) $k = 499$

Figure 9: LJL evaluated with a varying amount of nearest neighbors (500 points in total).

**The Impact of the Attraction Term**   We continue to discover the impact of attraction term in LJ potential. Two experiments, shown in Fig. 10 run in identical settings except for the appearance of the attraction term. Intuitively, the repulsive term alone should achieve an effect of redistribution. Without attraction term, however, particles are easily pulled out of the initial boundaries and spread out further. LJLs with the attraction term can enable the overall control of the point set and prevent particles from spreading out, which leads to a compact and equal distribution.

### A.2   REDISTRIBUTION OF POINT CLOUD OVER MESH SURFACES

As a motivating example for real applications, we use LJLs to evenly distribute points over mesh surfaces. Initially, the point cloud is randomly generated inside a 3D cube $[-1, 1]^3$. To ensure the generality of LJL parameters in 3D cases, meshes are normalized in the cube accordingly. The LJL parameters $\alpha = 0.5$, $\beta = 0.01$, $\epsilon = 2$ are the same as the previous example while $\sigma = 5\sigma'$ is due to the addition of the third dimension. In the following four cases illustrated in Fig 11, we show the importance of applying projection in-between LJL iterations. Without the help of LJLs, random points are directly projected on the sphere shown in Fig. 11(a). In the case where LJL is only used once as a pre-processing step before the projection, points are located on the surface but still non-uniformly distributed (see Fig. 11(b)). When using LJL as a post-processing after the projection, additional noise is introduced and points are shifted away from the surface shown in Fig. 11(c). Due to the fact that LJLs rearrange points in an iterative way, in the last case, we project points to the underlying surface in each intermediate LJL step. The resulting point configuration is not only evenly distributed, but also lies on the sphere shown in Fig. 11(d).

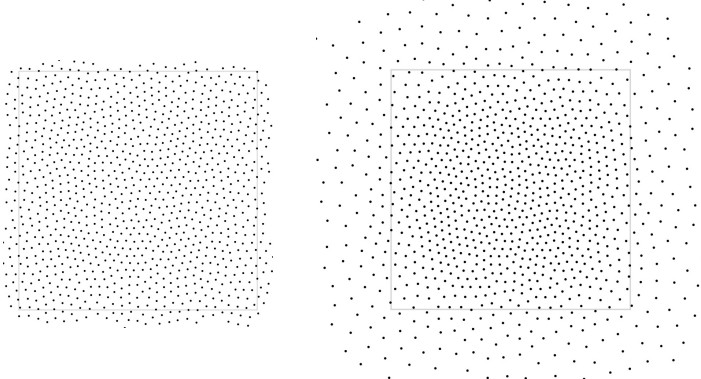

Figure 10: LJL with activated (left) and deactivated (right) attraction term.

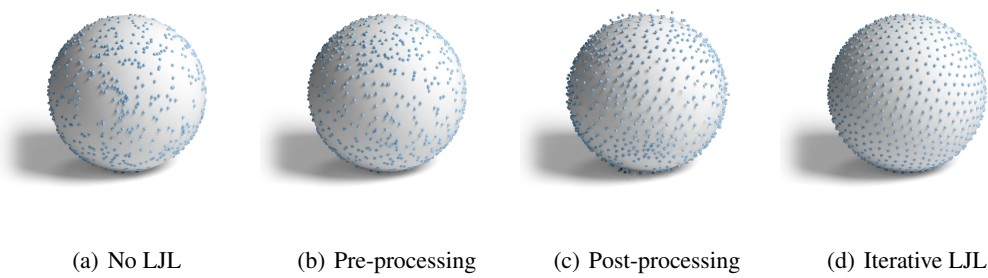

(a) No LJL            (b) Pre-processing            (c) Post-processing            (d) Iterative LJL

Figure 11: Redistribution of random 3D point cloud over a unit sphere. (a) Direct projection of random points. (b) Apply LJL before the projection as pre-processing. (c) Apply LJL after the projection as post-processing. (d) Apply projection in between LJL iterations.

In Fig. 12, two particles $A$ and $B$ form a cluster when they are directly projected on the surface ($A'$ and $B'$). With the help of LJL rearrangement, particles are moved to new positions ($A_{LJ}$ and $B_{LJ}$) and further projected back on the surface with proper distance ($A'_{LJ}$ and $B'_{LJ}$). In this case, the redistributed points are not only located on the surface but also have blue noise properties. The computations involved in LJL-guided point cloud redistribution are summarized in Algorithm 3, where we start with a random 3D point cloud $X_0$ initialized inside the cube $[-1, 1]^3$ and a normalized mesh surface $M$. In order to select nearest neighbors on the same side of the surface, LJLs only act on pairs that satisfy the condition that the intersection angle between their normals is less than $\frac{\pi}{4}$. In each LJL iteration, the redistributed points are projected back onto the surface. After a sufficient amount of LJL iterations, random point clouds can be rearranged to a normalized distribution and efficiently avoid the formation of clusters and holes.

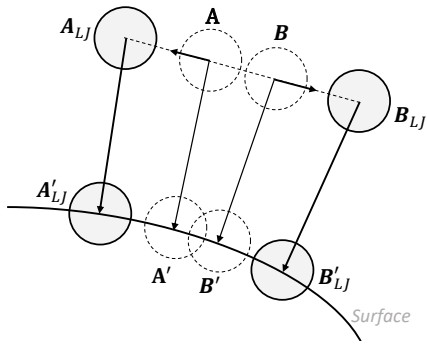

Figure 12: An illustration of how LJLs perform redistribution of clustered particles.

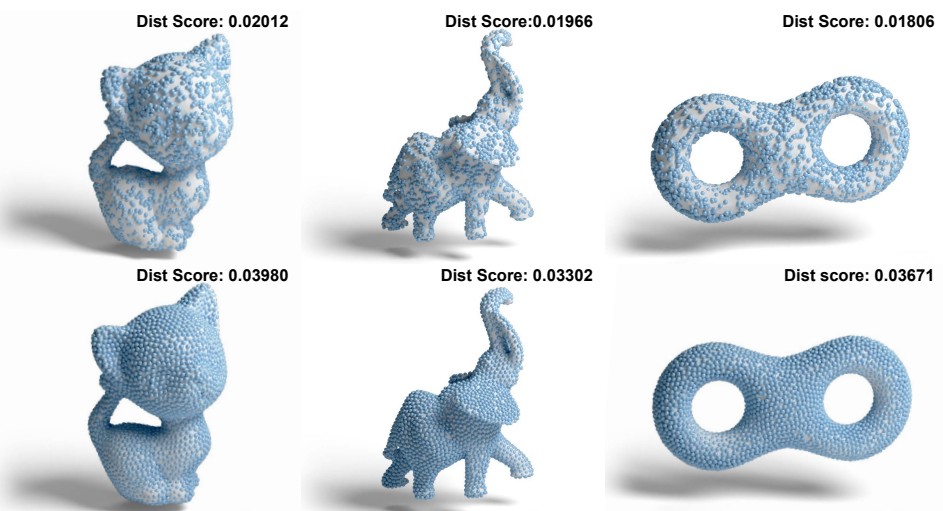

Figure 13: Projecting random 3D point clouds on mesh surfaces with (right) and without (left) LJLs (3000 sample points).

For geometric analysis, we use the mean value of Euclidean distances from each point to its nearest neighbor denoted as *distance score* to evaluate the point distribution of $N$ points shown in Equation 4. The nearest neighbors are those who satisfy the intersection angles of their normals less than $\frac{\pi}{4}$ to ensure they are on the same side of the surface. Higher *distance score* indicates the uniform point distribution and fewer holes and clusters in the results. The score is computed by

$$Distance\ score = \frac{1}{N} \sum_{i=0}^{N} |\boldsymbol{X_i} - \mathsf{NNS}_{\frac{\pi}{4}}(\boldsymbol{X_i})|. \tag{4}$$

The results of LJL-guided redistribution demonstrate significant improvements in point distribution, shown in Fig. 13.

---

**ALGORITHM 3:** *Distribution normalization of 3D point cloud over the mesh surface $\boldsymbol{M}$.*

**Input:** Point cloud $\boldsymbol{X}_0$ and tolerated threshold **tol**.
**Output:** Point cloud $\boldsymbol{X}_n$.

1    $n = 0$
2    **repeat**
3       $\boldsymbol{N}_n \leftarrow \boldsymbol{Normal}(\boldsymbol{X}_n)$
4       $\boldsymbol{N}_n' \leftarrow \boldsymbol{Normal}(\mathsf{NNS}(\boldsymbol{X}_n))$
5       **if** $\boldsymbol{Angle}(\boldsymbol{N}_n, \boldsymbol{N}_n') < \frac{\pi}{4}$ :
6         $\boldsymbol{X}_{n+1} \leftarrow \boldsymbol{LJL}(\boldsymbol{X}_n, \mathsf{NNS}(\boldsymbol{X}_n))$
7       **else**:
8         $\boldsymbol{X}_{n+1} \leftarrow \boldsymbol{X}_n$
9       $\boldsymbol{X}_{n+1} \leftarrow \boldsymbol{Project\text{-}to\text{-}Mesh}(\boldsymbol{X}_{n+1})$
10      $\Delta X \leftarrow |\boldsymbol{X}_{n+1} - \boldsymbol{X}_n|$
11      $n \leftarrow n + 1$
12    **until** $\Delta X <$ **tol**

---

## B LJL-EMBEDDED DEEP NEURAL NETWORKS

In this section, we show more details of LJL-embedded networks' parameter searching and results. We test all models on NVIDIA GeForce RTX 3090 GPU if not specified otherwise. Two well-trained point cloud generative models used in the papers are from ShapeGF (Cai et al., 2020) and DDPM (Luo & Hu, 2021b). The well-trained point cloud denoising model is from Luo & Hu (2021c).

### B.1 LJL-EMBEDDED GENERATIVE MODEL FOR 3D POINT CLOUD

In order to determine $\alpha$ and $\beta$, we perform a systematic parameter searching based on *distance* and *noise* scores, shown in Figure 14. We found that $\alpha = 2.5$ and $\beta = 0.01$ meet the requirements of less noise and better distribution. Assuming that it takes $T = 100$ steps to generate point clouds,

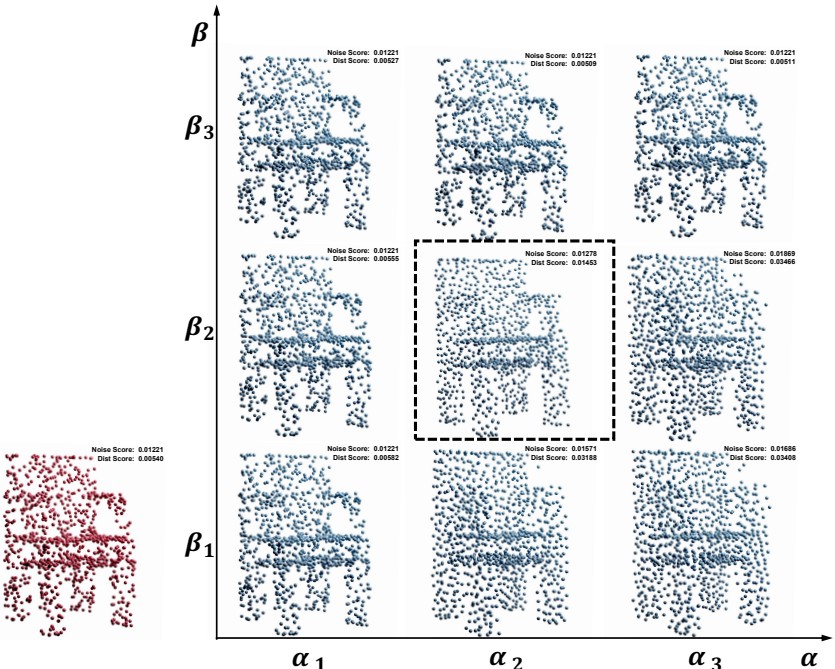

Figure 14: Parameter searching for decaying factors $\alpha$ and $\beta$, where $\alpha_1 = 0.1$, $\alpha_2 = 2.5$, $\alpha_3 = 5.0$ and $\beta_1 = 0.001$, $\beta_2 = 0.01$, $\beta_3 = 0.1$. The optimal parameters are $\alpha = 2.5$ and $\beta = 0.01$.

we continue to search for a good *starting steps SS* to insert LJLs shown in Fig. 16. $SS = 0$ means LJLs are applied from the very beginning, whereas $SS = 101$ means generation without LJLs. In order to balance the generation speed and LJL normalization effects, we choose $SS = 60$, meaning embedding should start in the second half of the generation steps.

We especially focus on the task of using as few points as possible in generation tasks. For use cases with a large number of points, inefficient point distributions are less of a concern since even for low-density regions, there are sufficient points to describe the underlying surface. Fig. 15 shows how LJLs behave in the generation process with different numbers of points $N$. Note that our choice of $\sigma = 5\sigma'$ implicitly accounts for the change in $N$. In the case of generation tasks with fewer points, the LJL-embedded generator can redistribute the limited number of points as even as possible while maintaining the global structure. Combining LJLs with generative models enables points to have the ability to converge to the different parts of the shape. More LJL-embedded generation results are shown in Fig. 17, Fig. 18, and Fig. 19.

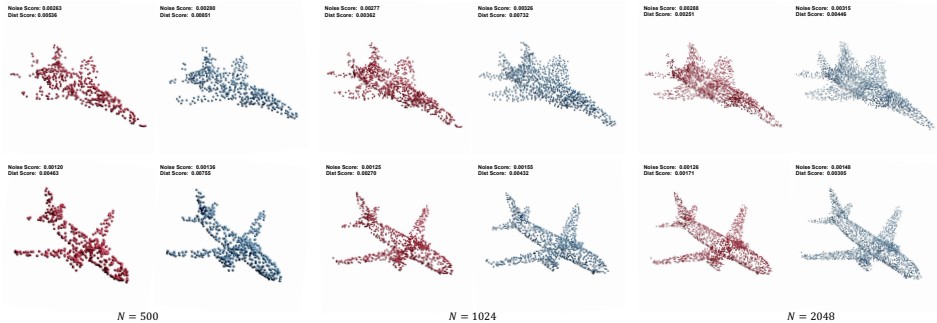

Figure 15: Examples of generation results with regard to the increasing number of sample points $N$. Red: Raw generation results. Blue: LJL-embedded generation results.

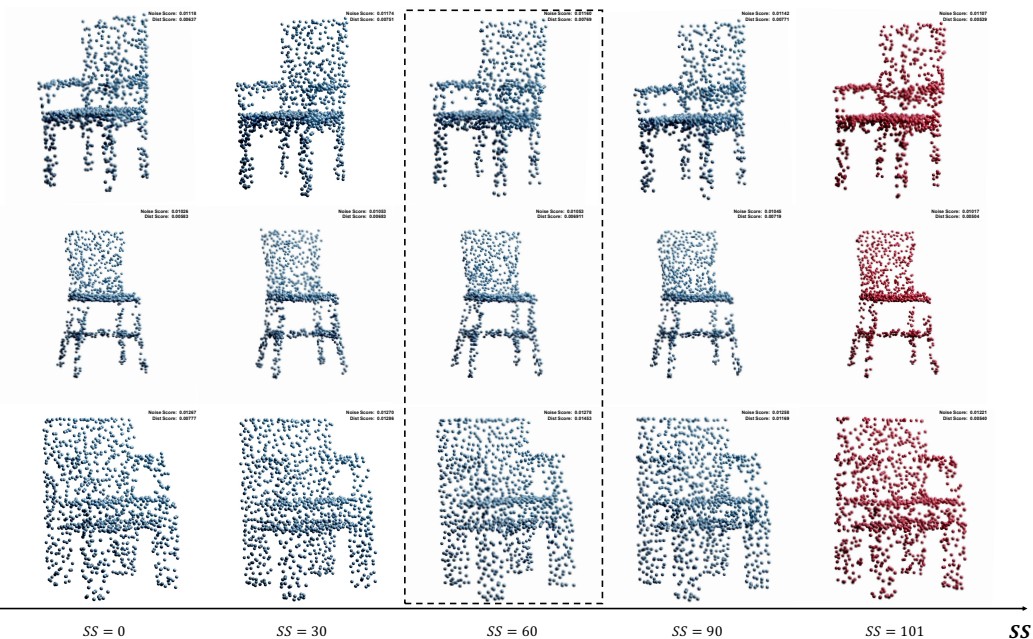

Figure 16: The optimal starting step ($SS$) to embed LJLs in the generation process (i.e. T=100). We select $SS = 60$ in the actual generation task.

## B.2 LJL-EMBEDDED DENOISING MODEL FOR 3D POINT CLOUD

LJL-embedded denoising results with different denoising iterations and noise scales are shown in Fig. 20 and Fig. 21.

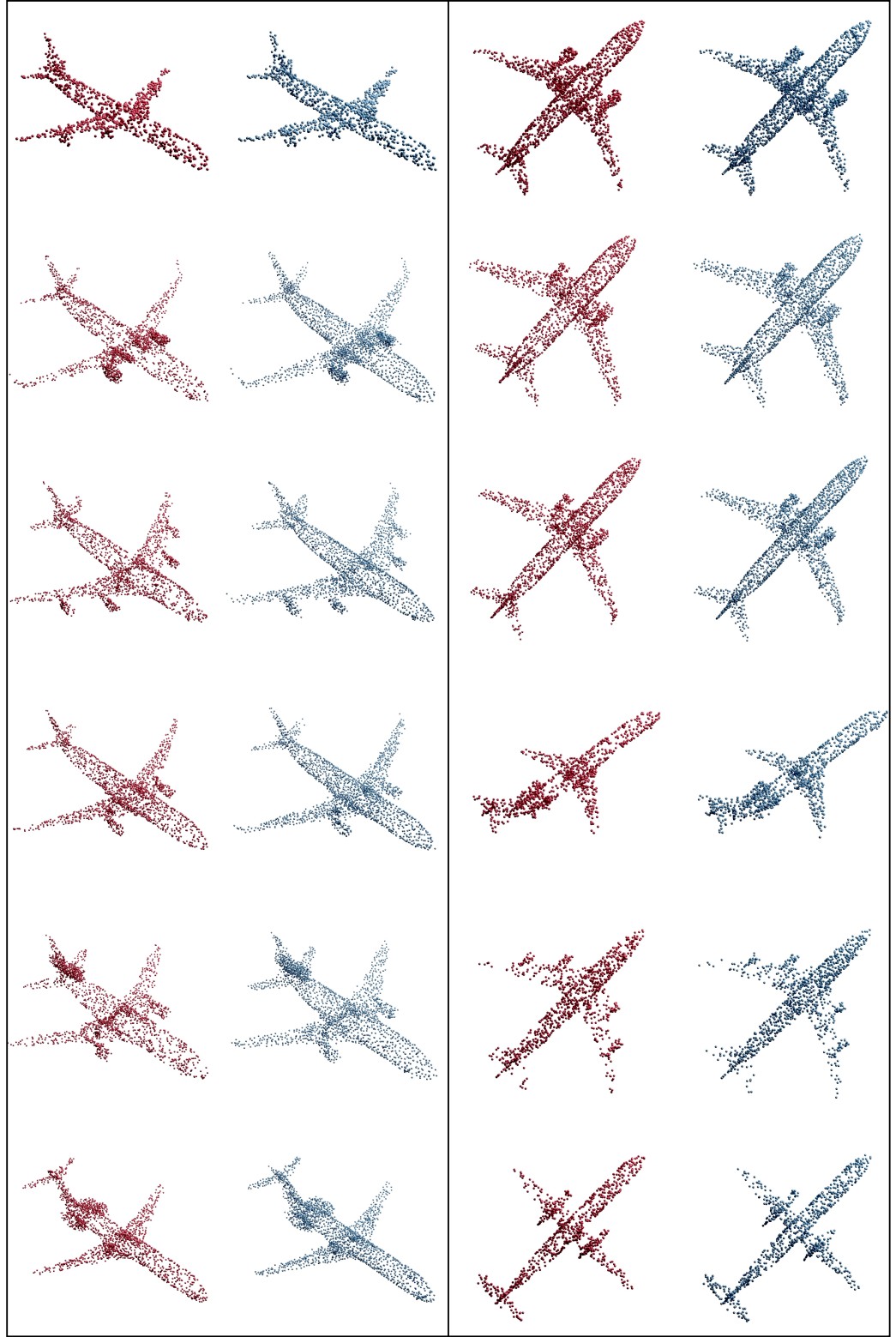

Figure 17: Comparison of raw generation (red) and LJL-embedded generation (blue).

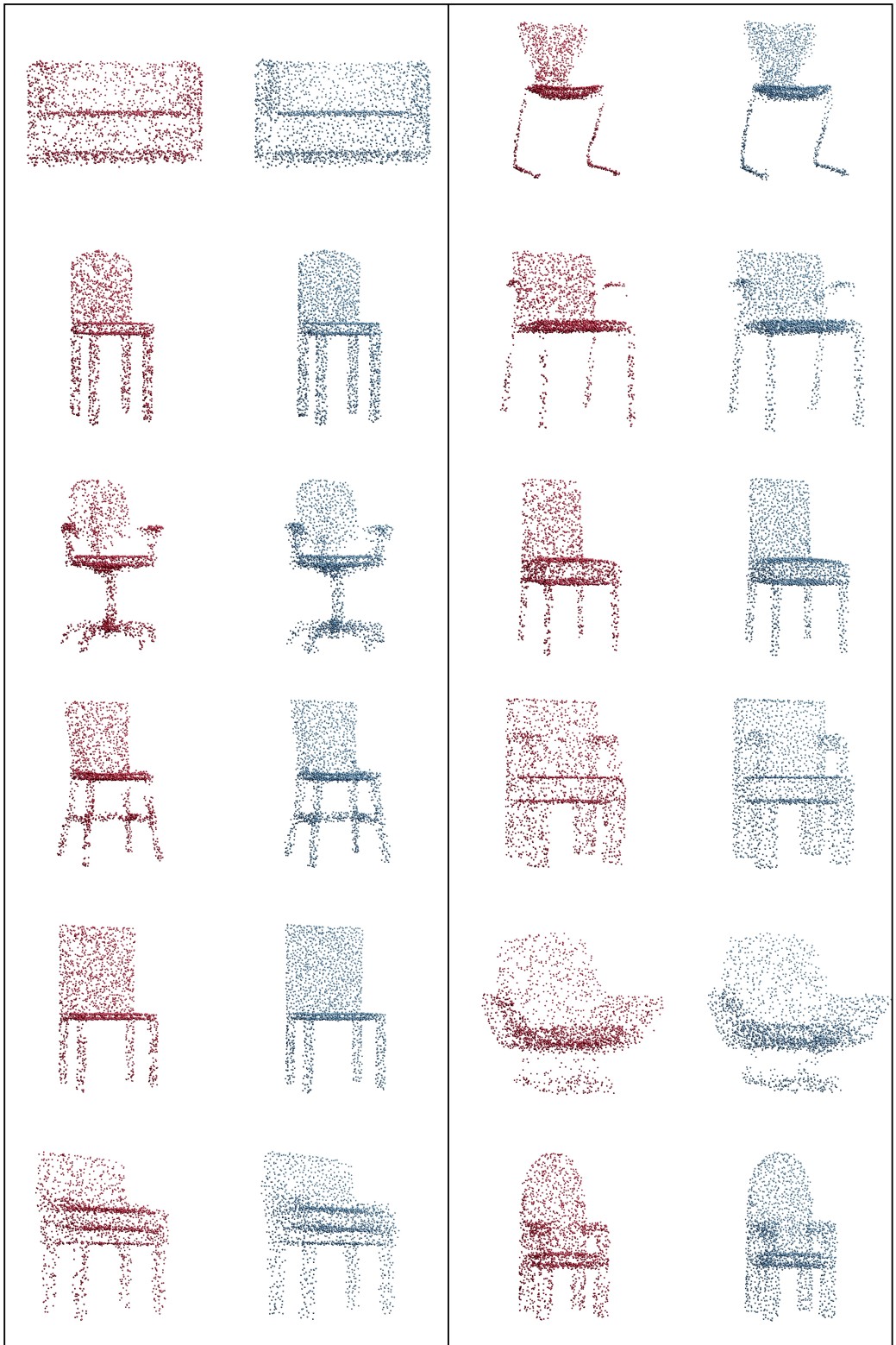

Figure 18: Comparison of raw generation (red) and LJL-embedded generation (blue).

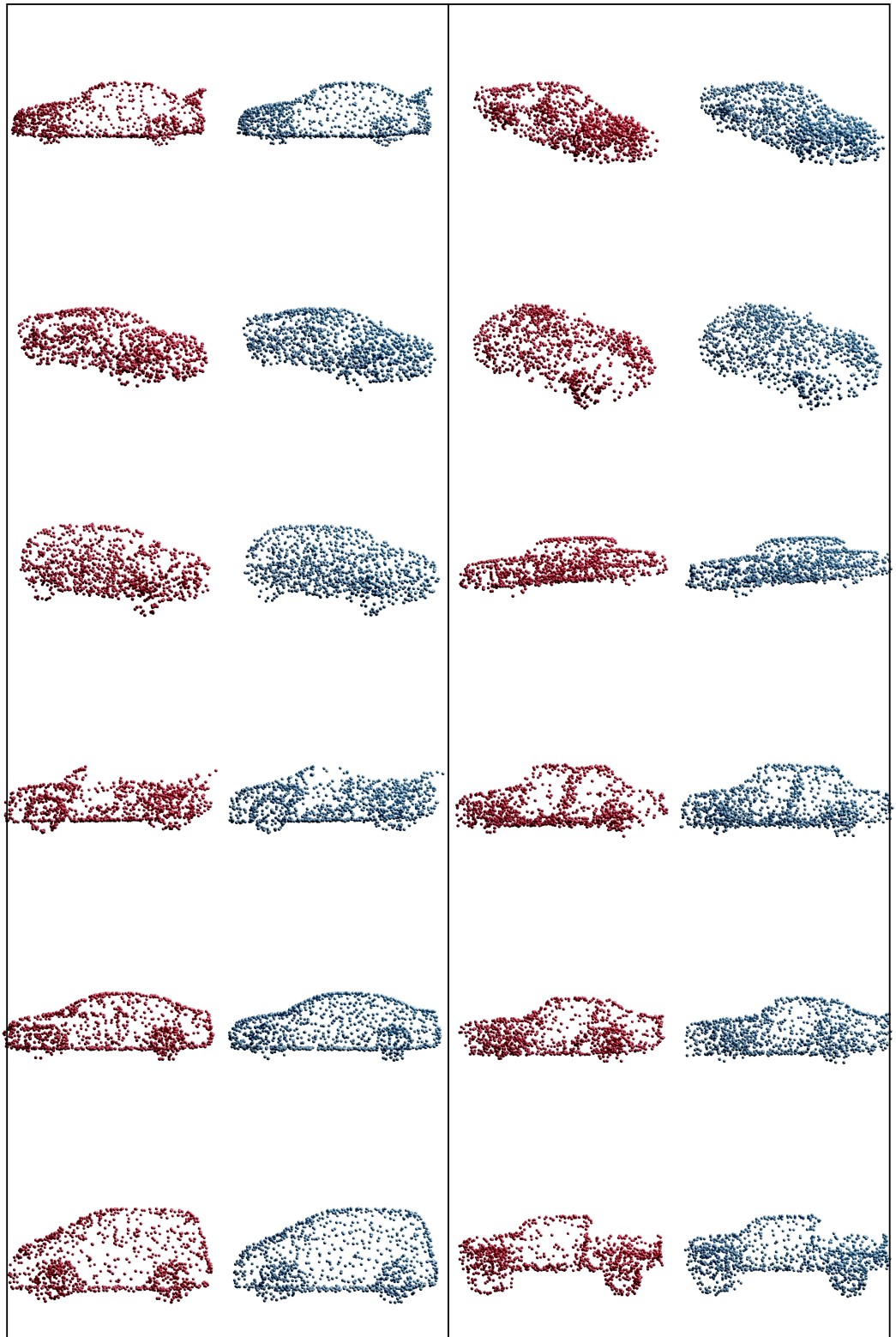

Figure 19: Comparison of raw generation (red) and LJL-embedded generation (blue).

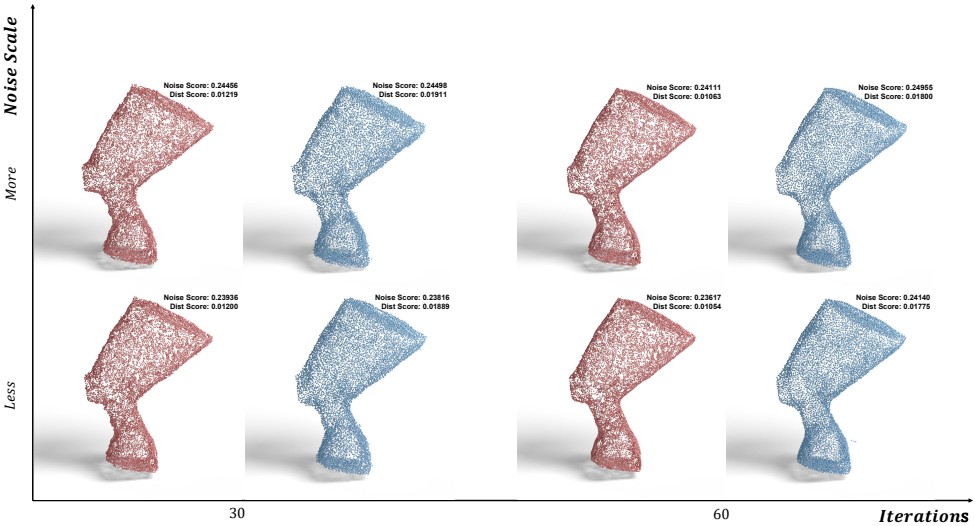

Figure 20: Comparison of denoise-only (red) and LJL-embedded denoising (blue) for 30/60 denoising iterations and different input noise scales.

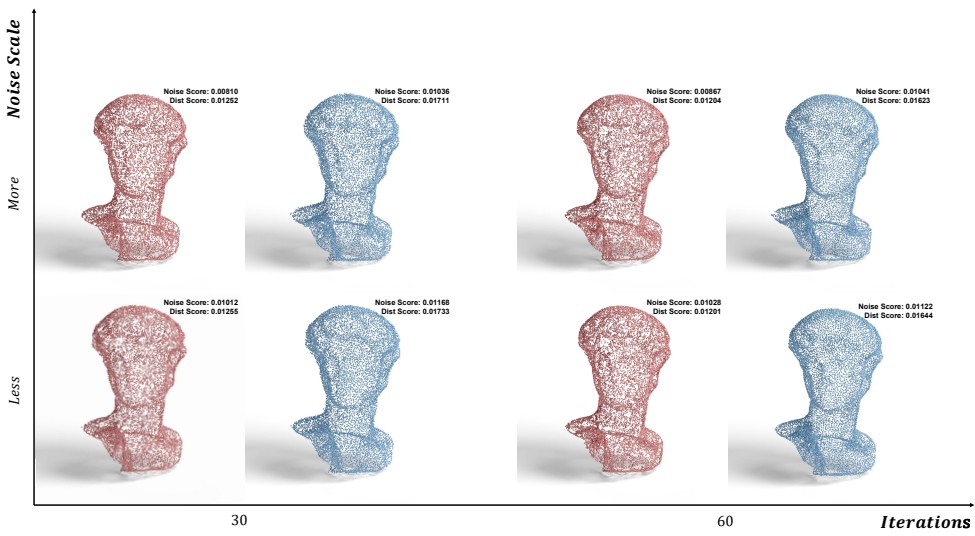

Figure 21: Comparison of denoise-only (red) and LJL-embedded denoising (blue) for 30/60 denoising iterations and different input noise scales.

