# OpenReview forum: "A Lennard-Jones Layer for Distribution Normalization"
_ICLR.cc/2024/Conference — ICLR 2024 Conference Withdrawn Submission_

### Official Review · Reviewer_tpha · 2023-10-29

**Soundness:** 3 good
**Presentation:** 3 good
**Contribution:** 2 fair
**Rating:** 3
**Confidence:** 4

**Summary:**

Learning-based 3D shape generation approaches, including auto-encoder, tend to generate 3D shapes with defects, i.e., shapes with holes and/or clusters, which is caused by inequal density distribution of points across the shape surface. To alleviate this issue, this paper proposed a Lennard-Jones layer (LJL) to equalize the density across the distribution of 2D and 3D point clouds while still keeping the overall shape structure. This process is termed as $\textit{distribution normalization}$.  Be more specific, Lennard-Jones potential is first computed for each pair of nearest points within a point cloud and each point is either pulled or pushed by the gradient of the potential as forces.

In addition to a toy example on 2D Euclidean plane, the proposed LJL is evaluated on auto-encoder-based generative model and DDPM-based generative model. The $\textit{Distance Score}$ proposed in this paper is used to evaluate the point distribution increase by a large margin when integrating LJL into the above two generative models, while the generation results are slightly affected。

**Strengths:**

1. $\textbf{Problem Formulation}$: This paper indeed identifies an issue with the existing learning-based point cloud generation models, i.e., the generated 3D shapes contain holes and/or clusters, which is undesirable.

2. $\textbf{Method soundness}$: In general, the adaptation of Lennard-Jones potential from chemistry and biology fields to redistribute 3D points is reasonable, as the formulation of LJ potential (Eq. 1) can push away points clustered together and pull points to fill up a hole.

3. $\textbf{Experimental Results}$: Both quantitative and qualitative results can validate the effectiveness of the proposed LJ layer in redistributing point distributions in generated point clouds.

**Weaknesses:**

1. $\textbf{Motivation}$

1.1 One concern is to what extend we need to redistribute points in generated point clouds? Holes and clusters indeed exist in the generated point clouds, but does this problem is severe enough? Some down-streaming or related tasks or applications which are severe affected by the inequal distribution of points are needed to strength the motivation of the paper.

1.2 If we generate more points for shapes and then uniformly downsample again, will the issue of holes and clusters be alleviated? This is related with the significance of the paper.

2. $\textbf{Method}$

2.1 According to Figure 3, it seems that the hyperparameters $\epsilon$ and $\sigma$ have significant influence on the redistribution result, so it may need case-by-case tuning of $\epsilon$ and $\sigma$ to achieve a good performance. Correct me if I'm wrong.

2.2 From Figure 5 and Figure 7, incorporating LJL could lead to over-smooth shape boundaries (wings and tailplanes in Figure 5) and slightly distortion of shape details (nose in Figure 7). This may be the drawback of the proposed approach. If such a drawback could be corrected, it will strengthen the paper.

3. $\textbf{Experiments}$

3.1 Some experimental details are missing. For example, how many points are generated per shape in experiments in Section 4.2? Did you retrain ShapeGF, and Lou & Hu's model or used their released pretrained model? Why not use the original evaluation metrics adopted in ShapeGF, and Lou & Hu's paper?

3.2 It is only evaluated on three generative approaches from two categories, e.g., audo-encoding and DDPM, more experiments and evaluations are needed.

4. $\textbf{Writting}$

4.1 The legend text in Figure 18 is too small to be seen clearly.

3.3 It is preferable to add citations in Table 1 and Table 2 to make it easier to check the reference papers.

**Questions:**

Please refer to the weakness part.

---

> ### Author Response · Authors · 2023-11-22
>
> 1. Motivation:
>
>
> We see the usefulness of our approach mainly in being able to reduce the number of points (thus increasing efficiency) while still expressing the surface well. For a huge amount of points, the surface is already known with high accuracy and explicit surface resampling can readily be used. However, the same is not true when a few points are used, since the surface to be resampled cannot be reconstructed accurately. As mentioned, we are adding another analysis of this.
>
> 2. Avoiding oversmoothing etc.:
>
>
> Since the LJL is task agnostic, it is always applied in the same way. Results with less surface distortion could probably achieved for each of the tasks when doing holistic optimization of this specific task, e.g. by training the network with already optimized distributions.
> In our work, we motivate the use of equally distributed point clouds and offer a general solution for this that does not require retraining of
> any networks. Improving the results further as stated above is then interesting future work, but outside the focus of this paper.

---

### Official Review · Reviewer_8NiC · 2023-11-01

**Soundness:** 3 good
**Presentation:** 2 fair
**Contribution:** 2 fair
**Rating:** 5
**Confidence:** 3

**Summary:**

This paper presents the Lennard-Jones layer, a plug-and-play layer to normalize point cloud distributions which can be added to ShapeGF and DDPM for point cloud generation.  The paper provides some analyses of the priorities of the layer and evaluates the proposed method on some toy examples then point cloud generation problem on ShapeNet.

**Strengths:**

- The proposed method inspired by Lennard-Jones potential is new for the distribution normalization of 3D point clouds. The solution looks reasonable and might be useful for future research in this area.

-  The paper is well-organized. It is nice to see the method is first evaluated using some toy examples and then extended to more complex cases.

**Weaknesses:**

Although some results presented in the paper look good, my main concern is that the paper fails to convincingly show the value or potential of the proposed method.

- The authors choose to use the 3D point cloud generation tasks to show the value of the proposed layer. However, I think it is still questionable whether the method can be generalized to other or more advanced point cloud generation methods beyond ShapeGF and DDPM. If the method is only compatible with these two relatively old methods, the contribution of this method might not be high.

- The irregular distributions of point clouds may carry useful information about the point clouds. Normalizing point cloud distributions may not always be helpful to improve generation results. To balance the irregular structures and normalized distributions, it may need prior knowledge to adjust the hyper-parameters of the proposed method. According to Figure 14, the method seems sensitive to these hyper-parameters. Do you have a systematic/automatic solution to determine these parameters? If it is difficult to determine these parameters or the parameters need to be determined case-by-case, the method may not be able to generalize to various problems.

- Table 1 and 2 only report the relative improvement. Can you provide detailed results? If the proposed method can directly improve the number reported in the original papers, the results will be more convincing.

**Questions:**

Please refer to my comments above.

---

> ### Author Response · Authors · 2023-11-22
>
> Generalization to other tasks:
>
>
> Similarly to our reply to Reviewer 1, we note, that LJL works for any iterative algorithm and is (except for the choice of parameters detailed in the general statement) task agnostic. In all studying tasks it works in the same way, there is no indication that that would not be the case for the next added case. Our choice of concrete networks was thus mainly motivated by code availability.

---

### Official Review · Reviewer_yVcr · 2023-11-02

**Soundness:** 3 good
**Presentation:** 3 good
**Contribution:** 3 good
**Rating:** 6
**Confidence:** 3

**Summary:**

This paper extends the concept of Lennard-Jones potential to describe the distribution of 2D and 3D point clouds, where points are regarded as particles with pairwise repulsive and weakly attractive interactions. Based on optimizing pair-wise Lennard-Jones potential, the whole point clouds could have better distribution. Applications in 3D point cloud generation and denoising tasks have proved the effectiveness that the proposed method is able to maintain uniform distribution of points.

**Strengths:**

1. The writing is well and the paper is easy to follow.
2. The idea is reasonable and solid due to the dependence on mechanism of real-world atoms and monocular, and the problem of distribution is pervasive in various situations, which might inspire other researchers in this domain.
3. The experiments show significant improvement in point distribution.

**Weaknesses:**

1.	Lack of clear guidance in choosing hyper parameters (eg. \alpha and \beta in Eq. 2). Although there’s discussion in Appendix, the authors still adopt grid search. Due to the diversity in various point clouds with different local/global density, it is better to provide a more clear guidance for choosing hyper parameters.
2.	The benefits of uniform distribution on downstream perception tasks (eg classification, segmentation, detection) is not verified, which is important since the application of point clouds mainly lies in perception tasks. This is not trivial since sometimes ununiform sampling is more effective (eg. Edge sampling in [1])
3.	All of these experiments are based on object-level point cloud. However, scene-level point cloud is more important in real-world applications. Is the proposed method still performs well when it comes to scene-level point clouds?
4.	The efficiency of this algorithm when processing large scale point clouds (~1 million points, which is common in some real-world datasets) is not mentioned.
[1]. Wu, Chengzhi, et al. "Attention-based Point Cloud Edge Sampling." In CVPR, 2023.

**Questions:**

All of my questions have been illustrated in Weaknesses.

---

> ### Author Response · Authors · 2023-11-22
>
> 1. Motivation:
>
>
> The motivation of distribution normalization is discussed in section 1. In short, we aim to have a higher expressiveness of few points by
> avoiding holes and clusters. We agree, that there are scenarios, where a more uniform distribution is not an advantage. Identifying whether or not distribution normalization is an advantage is the users' responsibility, analogous to how outlier removal or sorting of a data set is not always a useful operation
>
>
> 2. Scene-level point clouds:
>
>
> Our method operates locally, not globally (in fact, through the nearest neighbor search, the effect of each step is extremely locally limited). Since scenes are composed of individual objects and we sufficiently show individual objects with all kinds of surface features, it automatically follows, that LJL works on whole scenes as well. The local impact could however not be shown in large scenes, which makes the unsuitable for evaluating the effects of LJL. Thus, we did not include them.
>
> 3. Efficiency:
>
>
> The most complex step is the nearest neighbor search, which has a complexity of N*log(N). As long as the depth of the processing network is greater than log(N), the runtime is dominated by the original network, not the LJL. Even if this is not the case, there is no
> explosion of runtime.

---

> ### Comment · Reviewer_yVcr · 2023-12-03
> **Keep my rating**
>
> I appreciate the author's thorough response to my questions. After revisiting the paper and the rebuttal, I have decided to maintain my current rating.

---

### Official Review · Reviewer_bzSA · 2023-11-06

**Soundness:** 2 fair
**Presentation:** 2 fair
**Contribution:** 2 fair
**Rating:** 3
**Confidence:** 3

**Summary:**

The paper proposes incorporating Lennard-Jones potential into the problem of point cloud generation in order to obtain more uniformly distributed point clouds. Minimizing the LJ potential can be seen as moving the points so that close-by points are neither too far or too close to each other. This operation can be inserted into certain time steps of a point cloud diffusion model to prevent the final result from forming holes and clusters. The proposed method is benchmarked on 2D through spectral analysis, and 3D via a diffusion autoencoder experiment.

**Strengths:**

* The uniformity of generated point clouds is a problem that is under-studied. This paper might raise awareness of the problem and encourage future works.
* The connection drawn between uniformly distributed point clouds and blue noise is interesting.
* Being an algorithm that is extremely sensitive to hyperparameters, the effect of different hyperparameters are well ablated and visualized.

**Weaknesses:**

* The proposed method might only be useful for certain classes of point cloud generative models. The paper only demonstrates improved point cloud uniformity when used in conjunction with ShapeGF (Cai et al.) and DDPM (Luo et al.). However, these two models are inherently flawed -- they formulate point cloud generation as independent points uniformly distributed on the surface, thus unable to capture the global uniformity. Methods that models joint distribution of points, such as LION (Zheng et al.) and Point-E (Nichol et al.) might already produce uniform point clouds without the proposed method.
* Evaluation is lacking -- it will be more solid if the proposed method can be benchmarked against state-of-the-art models using standard point cloud generation and reconstruction evaluation metrics on standard datasets, instead of just presenting the percentage increase over a simple baseline.
* The exposition is sometimes comfusing. For example, it is not clear how the optimal LJL parameters are found using Algorithm 2 -- it shows merely a diffusion autoencoder for point clouds.
* Minor writting issue: In Algorithm 2, "autoencoder" usually refers to an encoder-decoder that is trained to reconstruct the input. In this case, it is better to call "E_\theta" as "encoder" instead.

**Questions:**

* In the paper, it seems that a fixed set of hyperparameters is used for all the shapes. Will it cause problem for 3D shapes with vastly different surface areas? Would it be better to tune the sigma values differently for different shapes?
* Could you elaborate on the connection between blue noise and the quality of point clouds?

---

> ### Author Response · Authors · 2023-11-22
>
> 1. Choice, which models to compare against:
>
>
> To the best of our knowledge, no other approach is specifically optimized to generate uniformly distributed point clouds. This includes LION and Point-E, where the training data does not contain uniform sampling and non-uniform sampling can be observed in the results shown in the paper. Based on the structure, LJL can be included in any iterative process. Our specific choice was motivated by the availability of source code of other methods. Since LJL works the same in every task, there is nothing that would indicate it would not work on a specific network like LION or Point-E.
>
> 2. Regarding the last question, blue noise is a type of randomized uniform sampling and we want to sample/generate/denoise point cloud to have this type of property.

---

> ### Comment · Reviewer_bzSA · 2023-11-22
> **Response**
>
> Thanks for the reponse.
> Could you also elaborate further on the relationship of sigma and surface area?
> I also wonder what are the use cases that require the point cloud to resemble blue noise? I think this will be an interesting avenue of research if you can show the effect of different point cloud distributions on downstream tasks.
>
> I would like to keep my original score as all the concerns remain.

---

> > ### Author Response · Authors · 2023-11-22
> >
> > 1. Because in all our experiments, the shapes are normalized into [-1,1]^3 cubic space, $\sigma$ relates to the number of points.
> > As we presented in the paper, section 4.1 for the 2D case $\sigma$ =$\sigma'$ and section 4.2 for the 3D case $\sigma$ =5$\sigma'$ due to the additional third dimension.
> >
> >
> > 2. The main reason is that we want to avoid clusters and holes in the point cloud, which will limit the point cloud expression, especially with few number of points. We've explored the downstream applications for LJL, such as 2D blue noise generation, 3D uniform sampling over mesh surfaces, 3D point cloud generation, and 3D point cloud denoising tasks.

---

### Author Response · Authors · 2023-11-22

Choosing Parameters
====================

We reiterate how parameters should be chosen and quote where these recommendations are explained in the paper. Since multiple reviewers had questions about this, we will make the recommendations more visible in the final paper.

4 Parameters need to be set:
- $\sigma$
- $\epsilon$
- $\alpha$
- $\beta$

The choice of parameters depends on the use case:
- 2D: All parameters are given in the paper in section 4.1.
- 3D: We perform an analysis on optimal $\sigma$ and $\epsilon$ in section A.2.
Users can use these values without further consideration.

$\alpha$ and $\beta$ are parameters related to the stepsize and thus need to be adjusted to the concrete task LJL is applied to. Note, that these adjustments only happen once in the beginning, afterward and model for a given task is processed with the same parameters.

$\alpha$ and $\beta$ parameters are found via a grid search algorithm as described in section B.1.

For the denoising network:
- $\alpha$: 0.3
- $\beta$: 0.01

For the generative network:
- $\alpha$: 2.5
- $\beta$: 0.01

Note, that these fixed parameters are possible since all models are scaled to the unique cube as pre-processing.


Evaluation Metrics
======================

Based on the suggestion of the reviewers, we are adding another evaluation: For a known ground truth surface, we generate an unevenly distributed point cloud and a point cloud with improved distribution as described in our method. From each of these point clouds, a surface is reconstructed and compared against the ground truth surface. Initial results show, that especially in the case of few point clouds, the reconstructed surface is closer to the ground truth surface, showing that LJL improves the accuracy. We will add the full evaluation to the final version.